# Different Geographic Strains of Dinoflagellate *Karlodinium veneficum* Host Highly Diverse Fungal Community and Potentially Serve as Possible Niche for Colonization of Fungal Endophytes

**DOI:** 10.3390/ijms24021672

**Published:** 2023-01-14

**Authors:** Yunyan Deng, Kui Wang, Zhangxi Hu, Qiang Hu, Yingzhong Tang

**Affiliations:** 1CAS Key Laboratory of Marine Ecology and Environmental Sciences, Institute of Oceanology, Chinese Academy of Sciences, Qingdao 266071, China; 2Laboratory of Marine Ecology and Environmental Science, Qingdao National Laboratory for Marine Science and Technology, Qingdao 266237, China; 3Center for Ocean Mega-Science, Chinese Academy of Sciences, Qingdao 266071, China; 4Institute for Advanced Study, Shenzhen University, Shenzhen 518060, China; 5College of Fisheries, Guangdong Ocean University, Zhanjiang 524088, China; 6Faculty of Synthetic Biology, CAS Key Laboratory of Quantitative Engineering Biology, Shenzhen Institute of Synthetic Biology, Shenzhen Institute of Advanced Technology, Chinese Academy of Sciences, Shenzhen 518055, China

**Keywords:** dinoflagellate-linked fungal community, endophytic fungi, hydrocarbonoclastic microorganisms, *Karlodinium*
*veneficum*, marine fungi, *Thyridium*, *Pseudeurotium*

## Abstract

In numerous studies, researchers have explored the interactions between fungi and their hosting biota in terrestrial systems, while much less attention has been paid to the counterpart interactions in aquatic, and particularly marine, ecosystems. Despite the growing recognition of the potential functions of fungi in structuring phytoplankton communities, the current insights were mostly derived from phytoplankton hosts, such as diatoms, green microalgae, and cyanobacteria. Dinoflagellates are the second most abundant group of phytoplankton in coastal marine ecosystems, and they are notorious for causing harmful algal blooms (HABs). In this study, we used high-throughput amplicon sequencing to capture global snapshots of specific fungal assemblages associated with laboratory-cultured marine dinoflagellate. We investigated a total of 13 clonal cultures of the dinoflagellate *Karlodinium veneficum* that were previously isolated from 5 geographic origins and have been maintained in our laboratory from several months to more than 14 years. The total recovered fungal microbiome, which consisted of 349 ASVs (amplicon sequencing variants, sequences clustered at a 100% sequence identity), could be assigned to 4 phyla, 18 classes, 37 orders, 65 families, 97 genera, and 131 species. The fungal consortium displayed high diversity and was dominated by filamentous fungi and ascomycetous and basidiomycetous yeasts. A core set of three genera among all the detected fungi was constitutively present in the *K. veneficum* strains isolated from geographically distant regions, with the top two most abundant genera, *Thyridium* and *Pseudeurotium*, capable of using hydrocarbons as the sole or major source of carbon and energy. In addition, fungal taxa previously documented as endophytes in other hosts were also found in all tested strains of *K. veneficum*. Because host–endophyte interactions are highly variable and strongly case-dependent, these fungal taxa were not necessarily genuine endosymbionts of *K. veneficum*; instead, it raised the possibility that dinoflagellates could potentially serve as an alternative ecological niche for the colonization of fungal endophytes. Our findings lay the foundation for further investigations into the potential roles or functions of fungi in the regulation of the growth dynamics and HABs of marine dinoflagellates in the field.

## 1. Introduction

Phytoplankton are essential components of marine ecosystems due to their critical roles in primary production, matter cycling, energy flow, and other biological and geochemical processes. The highly heterogeneous assemblages of phytoplankton grow in close association with microbial communities with immense and complex diversity [1]. While microalgal cells capture solar energy and transform inorganic matter into organic matter, they release various compounds into their immediate milieus, forming micro-spaces that are known as phycospheres, which are rich in organic molecules and can be viewed as the aquatic analogues of rhizospheres [2]. The phycosphere provides a niche for the colonization of heterotrophic microbes, including bacteria, fungi, and other protists. Generally, phytoplankton supply oxygen and organic nutrients to microbial communities, whereas the latter decompose organic matter, recycle nutrients, and contribute to the ecosystem balance [2,3]. According to mounting evidence, the mutual interactions between phytoplankton and microbiota act as vital factors that influence the dynamics of both algal populations and microbial communities [3,4,5,6,7,8]. However, compared with numerous studies on the prokaryotic association with phytoplankton, investigations on the coexistence of eukaryotic consortia with marine microalgae are much less addressed in the literature.

Fungi are persistent in marine ecosystems; however, they are understudied compared with their terrestrial counterparts [9,10]. As an integral component of marine microbiomes, fungi are ubiquitously found in nearly every marine habitat, from coastal areas to the deep ocean [9]. Although considerable efforts have been made to explore fungus–biota interactions in terrestrial systems, much less endeavor has been undertaken to elucidate the paralleled interactions in aquatic, and particularly, marine environments [9,10,11]. In a few field studies in open-ocean and coastal ecosystems, the authors demonstrated positive correlations between phytoplankton and fungal abundance [4,5,6,7,12,13,14,15,16]. Early diverging zoosporic fungi, such as Chytridiomycota, which are often referred to as chytrids, are frequently reported to be virulent parasites in a number of phytoplankton species, causing changes in their distributions and population successions [4,17]. In addition, the biotic relationships of marine fungi may go beyond direct interactions. According to several lines of evidence, marine fungi can assimilate and decompose substantial amounts of phytoplankton-derived organic matter, influence phytoplankton population dynamics, and even act as trophic links between phytoplankton and zooplankton via the mycoloop, which researchers initially described in freshwater ecosystems [4,9,10,15,16,17,18]. Despite a growing body of evidence indicating the presence and prevalence of certain fungal species in association with phytoplankton, thus far, these relationships in marine ecosystems have been primarily explored at the population and/or community levels; however, the nature of these interactions remains unclear [9,11]. The association of chytrids with phytoplankton is probably one of the most prominent examples for which the discrimination of mutualistic, saprophytic, or parasitic relationships is difficult [15]. Therefore, the nature of the interaction between the phytoplankton and co-occurring fungi in marine environments is still unknown and calls for more intensive studies from the fundamental aspects.

Dinoflagellates are the second most abundant phytoplankton group in coastal marine ecosystems. They are nontrivial contributors to total global carbon fixation; however, they are also infamous as chief agents of harmful algal blooms (HABs) [19]. In recent years, we have seen growing recognition of the potential functions and roles of fungi in structuring phytoplankton communities and trophic linkages in marine ecosystems; hitherto, the vast majority of the foregoing insights were derived from studies focusing on phytoplankton hosts such as diatoms, green microalgae, and cyanobacteria [4,6,9,10,15,18]. In quite a few case studies, the authors focused on the seasonal fungal abundance linked with the dinoflagellate population and the fungal dynamic patterns during dinoflagellate blooms. For instance, according to time-series data, the seasonal patterns of mycoplankton populations are tied to environmental gradients and phytoplankton communities, including dinoflagellates [5,6,7,13]. In another field study, the authors demonstrated that the diversity and composition of fungal communities appear to be relevant to phytoplankton dynamics during the bloom of the dinoflagellate *Noctiluca scintillans* [20]. Basidiomycota and Ascomycota were the primary fungal members detected in a dinoflagellate *Alexandrium catenella* bloom, and their abundance increased during the bloom terminal stage, which implies that there might be a saprophytic association between fungi and the decomposition of the *A. catenella* biomass [12]. The metagenomic sequencing of environmental samples (especially from algal blooms) has so far provided valuable insights to enrich our understanding of algal–fungal associations in general; however, it fails to determine which fungal taxa are associated with which specific dinoflagellate species, as other mycoplankton in seawater, could be misidentified as algal-associated. Meanwhile, the available information on the fungal associates of laboratory-cultured marine dinoflagellate species is still restricted to a few isolated reports [21,22]. Therefore, the association/interaction of fungal communities with hosting dinoflagellates, both in the marine environment and laboratory cultures, currently remains a “gray box.”

Unarmored dinoflagellate *Karlodinium veneficum* is a cosmopolitan species [23] that commonly forms HABs worldwide (as reviewed in [24]). Due to its ability to produce karlotoxins [25], its blooms usually cause massive mortality in fish, mussels, and zooplankton [24,26,27]. Given the frequency of *K. veneficum* blooms, their severe impacts on aquaculture and public health, and environmental concerns, the species has drawn substantial attention and has been intensively studied from multiple aspects. However, it remains an open question as to whether this common HAB-forming dinoflagellate is also associated with its specific fungal community. To better understand the possible contribution of fungal taxa to dinoflagellate bloom dynamics, it is necessary to compare fungal community persistence in laboratory cultures and consortia that coexist with natural populations in the field. In this study, our aim was to capture global snapshots of fungal assemblages associated with cultured marine dinoflagellate. We investigated the fungal communities that coexist with 13 clonal cultures of *K. veneficum* that have been isolated from 5 geographic origins (from the north to south coastal waters in China and an estuary of Chesapeake Bay in the United States) via high-throughput gene amplicon sequencing for the rDNA ITS (internal transcribed spacer) region. Our work provides the first global snapshot of the specific association/interaction of fungal communities with host marine dinoflagellate species and raises the possibility that dinoflagellates could potentially serve as an alternative ecological niche for the colonization of fungal endophytes.

## 2. Results

### 2.1. Global Overview of Fungal Communities Coexisting with K. veneficum

A total of 1,072,979 raw rDNA sequence reads from the 13 samples corresponded to 0.52 Gb of raw data, with the sequences per sample ranging from 80,234 to 85,511 (Appendix A). The raw sequencing data were deposited in the NCBI Short Read Archive (SRA) database with accession number PRJNA848294. The goods coverage value (an indicator of the sample completeness) of all the samples was 1.00 (Appendix A), and all the rarefaction curves tended to reach saturation with the increased sequencing amounts (Appendix A), which suggests that we harvested a sufficient number of sequences to uncover the majority of the taxa in all fungal assemblages. The denoising of the raw reads using the DADA2 plugin within the QIIME 2 tool resulted in 1,019,982 effective sequences, which were then binned into 349 amplicon sequencing variants (ASVs, sequences clustered at a 100% sequence identity). The number of ASVs per sample varied from 23 to 120 (mean = 44) (Appendix A). Based on the assignments in the UNITE database, all the obtained ASVs belong to the fungi kingdom and are thus used for subsequent analyses.

The whole fungal microbiome, which consisted of 349 ASVs, was composed of Ascomycota (91.40%), Basidiomycota (7.12%), and unclassified (UC) at the phylum level (1.40%), with the rare occurrence of Chytridiomycota (0.08%) (Figure 1A). The ASVs within these 4 phyla were further classified into 18 classes, 37 orders, 65 families, 97 genera, and 131 species (Appendix A). At the class level, Sordariomycetes (47.70%), Dothideomycetes (22.73%), and Saccharomycetes (8.98%) within the phylum Ascomycota exhibited relatively high abundance, ranking among the top three predominant members (Appendix A). At the genus level, we identified a set of 15 genera as “abundant,” with relative abundance greater than 1% in all 13 samples collectively, which together contributed to up to 83.06% of the entire fungal community (Figure 1B). We hereafter refer to these 15 genera as “abundant genera.” Among them, *Thyridium* was the predominant genus (36.40%), with *Pseudeurotium* as subdominant (11.54%), followed by *Candida* (8.23%); *Cladosporium* (3.89%); unclassified Ascomycota (3.89%); *Arthrinium* (3.31%); *Tetracladium* (2.59%); *Cryptococcus* (2.54%); *Malassezia* (2.17%); *Fusarium* (1.95%); *Acremonium* (1.79%); yet to be described “UC Fungi” (1.40%); *Phialocephala* (1.18%); *Teratosphaeria* (1.09%); *Thermomyces* (1.08%) (Figure 1B and Appendix A).

### 2.2. Species Diversity and Composition of Fungal Communities Coexisting with K. veneficum

Ascomycota was the most dominant fungal phylum, representing 79.48–99.77% of the relative abundance among the 13 KV strains (Figure 2A). Basidiomycota was the second most dominant fungi in almost all samples except for KV15 (more UC fungi present). Chytridiomycota made up merely 1.02% of the relative sequence abundance in KV17 (Figure 2A). The majority of the Ascomycota ASVs were phylogenetically assigned to Sordariomycetes, with a few exceptions, which were more represented by Dothideomycetes (Appendix A). Most of the Basidiomycota ASVs were affiliated with Tremellomycetes, although a few samples were more represented by UC Ustilaginomycotina (Appendix A). At the genus level, three members within Ascomycota, *Thyridium*, *Pseudeurotium*, and *Candida*, were found persistent in lab cultures of all 13 KV strains with a minimum relative abundance greater than 0.1% (Figure 2B and Table 1). Hence, they are considered the core genera of the fungal microbiome coexisting with *K. veneficum* (hereafter referred to as “the core genera”). The set of 15 abundant genera was found predominant in 12 out of the 13 samples (except for KV17), which altogether accounted for 66.95–99.99% of the relative abundance (Figure 2B, and Table 1). The PCoA plot illustrated that most of the fungal communities in the KV cultures did not closely cluster together based on their original host geographic sources at the ASV level (Figure 3). Nonetheless, 3 samples, KV6, KV7, and KV8, all from coastal waters in Shandong province, China, grouped together to form a distinct cluster, whereas the remaining 10 fungal communities exhibited indiscernible affiliations with one another (Figure 3).

### 2.3. Functional Predictions of the Whole Fungal Microbiome of K. veneficum

To primarily explore the metabolic potentials of the entire fungal microbiome in the associated KV phycospheres, the functional profiles of fungal consortia were inferred from the MetaCyc metabolic pathway database. Overall, the entire fungal microbiome performed the broad functions of “biosynthesis” (61.52%), “generation of precursor metabolites and energy” (21.49%), and “degradation/utilization/assimilation” (16.99%) (Figure 4). At the further level of classification, the terms “nucleoside and nucleotide biosynthesis” (28.98%), “amino acid biosynthesis” (8.20%), “fatty acid and lipid biosynthesis” (7.85%), “cofactor, carrier, and vitamin biosynthesis” (7.65%), “respiration” (6.37%), and “carbohydrate biosynthesis” (5.00%) displayed relatively high abundance (≥5%), ranking among the top predominant functional modules (Figure 4).

## 3. Discussion

### 3.1. K. veneficum Could Foster Diverse Fungi to Thrive in Cocultures after Original Isolation

Although some case studies have attempted to characterize certain lineages of fungi that associate and/or interact with marine dinoflagellates in laboratory cultures [21,22], the current knowledge is impeded due to biases in culture-based approaches and/or varying consistency and the persistence of fungal communities across different strains. In this study, we first used high-throughput amplicon sequencing to capture global snapshots of specific fungal associations with host marine dinoflagellates. A total of 349 fungal ASVs were retrieved from 13 *K. veneficum* strains originating from five different geographic origins. One may suspect that the fungal taxa detected in the algal cultures were contaminants of fungi that existed in the air milieu of the laboratory. We addressed this concern from two facets. First, we primarily drew the conclusions in this study from the analyses of 15 abundant fungal genera (i.e., all exhibiting a relative abundance greater than 1% in all 13 samples, together accounting for up to 83.06% of the entire fungal community). Of these 15 abundant genera, apart from two members (unclassified Ascomycota and unclassified Fungi) that have yet to be classified at the genus level (not feasible for the document search), all the other 13 genera, as a matter of fact, have been previously reported from marine environments (Appendix A), with 7 of them (*Candida*; *Cladosporium*; *Cryptococcus*; *Malassezia*; *Arthrinium*; *Fusarium*; *Acremonium*) recorded as common members in marine habitats (summarized in Appendix A). Moreover, all of these fungal taxa detected in a certain strain of laboratory-cultured *K. veneficum* have also been previously reported to be present in the adjacent seawater(s) where the *K. veneficum* strain was initially established (see Appendix A for more details). Second, since the 13 *K. veneficum* cultures employed in this study were isolated from different localities and maintained for different lengths of time, these cultures could be viewed as mutual controls of one another, somewhat equivalent to the time-series sampling for the field samples. Only 3 of the 15 abundant genera (*Thyridium*, *Pseudeurotium*, and *Candida*) were detected in all of the 13 *K. veneficum* cultures. This means that if the detected fungal taxa were laboratory contaminants, many more than three genera would be detected in all 13 cultures. Furthermore, we also conducted the same high-throughput gene amplicon sequencing of the rDNA ITS region for another 5 cultures of microalgae that functioned as controls. All 18 cultures (the 13 *K. veneficum* and the 5 control cultures) were cultivated and processed using the same protocols in our laboratory. It turned out that the abovementioned 3 fungal genera were not present in all 18 cultures (see Appendix A for more details). Taken together, the abovementioned evidence suggests that the vast majority of the fungal taxa that we identified in this study were not laboratory contaminants but rather were carried over from the seawaters in which the 13 *K. veneficum* cultures were initially established and have been carried along with numerous culture transfers.

The entire fungal community that coexists with *K. veneficum* is characterized by the ubiquitous dominance of the Dikarya subkingdom (including Ascomycota and Basidiomycota), as well as the episodic occurrence of zoosporic Chytridiomycota at a small percentage. This finding is consistent with the general notion that Dikarya members are dominant fungal types that are ubiquitously found in marine coastal sites [7,10,11,14,16,20]. The set of 15 abundant genera was composed of ten filamentous fungi (*Thyridium*; *Pseudeurotium*; *Cladosporium*; *Arthrinium*; *Tetracladium*; *Fusarium*; *Acremonium*; *Phialocephala*; *Teratosphaeria*; *Thermomyces*) and three yeasts (*Cryptococcus*; *Malassezia*; *Candida*), which indicates that filamentous fungi and ascomycetous and basidiomycetous yeasts were the predominant players in the whole consortia. The entire dataset provided an extraordinary diversity of fungal assemblages associated with marine dinoflagellate species, covering cosmopolitan, cold-adapted, and thermophilic taxa, which have been reported as saprobic, mutualistic, and parasitic species (see below). *Pseudeurotium*, *Fusarium*, *Arthrinium*, *Tetracladium*, *Phialocephala*, *Candida*, *Cladosporium*, *Cryptococcus*, and *Malassezia* are now known to have global distributions [15,28,29,30]. The *Pseudeurotium*, *Cryptococcus*, *Cladosporium*, *Acremonium*, *Teratosphaeria*, and *Phialocephala* genera are psychrophiles or psychrotrophs that adapt to or live in cold environments [31,32]. The *Thermomyces* genus comprises thermophilic members with optimum growth temperatures of 50 °C or higher [33]. The retrieved fungal community included members who played roles as saprobes, symbionts, parasites, and pathogens. For example, *Candida* is a large genus that consists of saprotrophic yeast [34]. Many *Cladosporium* species are plant pathogens [34], whereas *Malassezia* is a common parasite in marine animals [29] and may live as symbionts with sponges [35]. *Phialocephala* live pathogenic and saprophytic lifestyles [28]. Furthermore, from the predicted metabolic perspective, all fungal assemblages could conduct metabolic processes that involve carbohydrates, amino acids, energy, fatty acids and lipids, nucleosides, nucleotides, and other compounds, such as vitamins, secondary metabolites, etc.; however, this still requires further experimental validation. Together, these findings imply that the dinoflagellate *K. veneficum* recruits a high diversity of fungal communities that are capable of broad and diverse functional potentials, which have yet to be fully described.

### 3.2. K. veneficum Constitutively Attracts a Set of Hydrocarbonoclastic Microorganisms in Phycospheres

In this study, fungal taxa that were commonly and abundantly present across different samples were regarded as the core or stable constituents of the dinoflagellate-associated microbiome, which corresponds to the definition adopted in previous studies on dinoflagellates (as reviewed in [36]). Although the different strains of *K. veneficum* exhibited highly heterogeneous ASV compositions, as illustrated in the PCoA analyses, three core genera, *Thyridium*, *Pseudeurotium*, and *Candida*, were persistent in their phycospheres, together accounting for 56.21% of the whole fungal microbiome. The predominant *Thyridium*, formerly known as *Phialemoniopsis* [37] and *Phialemonium* [38], is a newly described Sordariomycetes genus. This genus has been found in soil [39], clinical samples [39], seawater samples [40,41], marine sediment [42], and a deep-sea hydrothermal vent [43]. Members of this genus have been isolated from chronically petroleum-contaminated seawater [40], and are known to grow on biodiesel as the sole carbon source in soil [39]. The putative enzymes and pathways that are potentially linked to diesel fuel degradation were also detected in the genome of *Thyridium curvata* (*Phialemoniopsis curvata*) [39]. Together, the evidence strongly implies that the members of the *Thyridium* genus are diesel degraders that are capable of using hydrocarbons as the sole or major source of carbon and energy. The second abundant core genus, *Pseudeurotium*, contains endophytic fungi that have a cosmopolitan distribution [31]. Some members of this genus grow on volatile aromatic hydrocarbons as their sole carbon and energy sources [44,45] and survive in high concentrations of diesel fuel in soil [46], which indicates their hydrocarbon utilization capacity. We recently found three genera of hydrocarbon-degrading bacteria, including two highly specialized clades, which live in stable co-occurrence with *K. veneficum* [36]. Collectively, these findings suggest that *K. veneficum* could constitutively attract a suite of hydrocarbonoclastic microorganisms in phycospheres. Although it is yet unclear what exact roles these hydrocarbon degraders played during their association with *K. veneficum*, the phycosphere of *K. veneficum*, or even certain lineages of dinoflagellates (e.g., Kareniaceae), seems to represent a previously unrecognized “hot spot” attractive to microorganisms that possess specialized nutritional preferences for hydrocarbons. Considering the long-term cultivation period (from several months to over 14 years) in laboratory cultures, such stable co-existence could not be explained by simple stochastic associations, but instead, a relationship that is beneficial to at least one party.

### 3.3. Dinoflagellates Potentially Serve as Alternative Niche for Colonization of Fungal Endophytes

With research progress regarding endophytic microorganisms, fungal endophytes have historically held several definitions. Currently, the most accepted definition for established endophytic fungi is fungi that entirely reside within plant tissues without causing apparent disease symptoms ([47] and the references therein). Over the past two decades, there has been exponential growth in the research on this topic in terrestrial plants [47], whereas limited attention has been paid to fungal plant endophytes in marine habitats. Hitherto, the vast majority of current insights into the endophytic fungi of marine algae were derived from macroalgae (see below). In this work, fungal taxa previously documented as endophytes in other hosts were prevalent in all the *K. veneficum* cultures. Of these abundant genera, *Cladosporium*, *Acremonium*, *Arthrinium*, and *Fusarium* are among the most common endosymbionts of seagrasses [30,48], as well as taxonomically unrelated and geographically disparate marine macroalgae [30,46,49,50,51,52]. The genus *Tetracladium* contains ubiquitous root endophytic aquatic hyphomycetes, which also exist as root endophytes in terrestrial environments [53]. *Pseudeurotium* has been reported as an endophyte in the green alga *Ulva lactuca* [46] and shrub *Rhododendron tomentosum* [54]. *Phialocephala* is one of the most common and widespread root endophytes of land plants [29]. *Thermomyces* is a thermophilic endophyte of desert plants [34,55]. *Candida* occurs as endophytic yeast in *Rosa canina* [56], *Prunus domestica* [56], and *Orchis tridentate* [57]. *Cryptococcu* is one of the more common endophytic yeast genera, and it can be isolated from coastal grasses and the fruits, leaves, stems, and roots of terrestrial plants [56,57,58,59,60]. *Teratosphaeria* is among the poorly understood root-colonizing groups of terrestrial plants [61,62,63].

Plant-infecting microorganisms can generally be classified as either pathogens or mutualists [64,65]. However, in reality, the situation is far more complex. The net effects of infection on the host are highly conditional and can virtually shift from mutualism to antagonism for any type of plant–microbe interaction [65,66]. The term “endophyte” does not describe a fixed life history trait but refers to a specific microorganism in a specific host under a specific set of environmental conditions [64,65,66]. Therefore, in our study, the detected fungal taxa previously documented as endophytes in other hosts were not necessarily living in endosymbiosis with *K. veneficum*, but rather, it suggested that dinoflagellates might possibly serve as an alternative niche for the colonization of fungal endophytes. To verify this hypothesis and gain further insights into the subject, techniques like the fluorescence/confocal microscopy with specific staining, and CARD-FISH with specific probes will certainly be applied to our future studies. A detailed investigation combining molecular assessment with three-dimensional confocal laser scanning and electron microscopy would obtain a more comprehensive understanding of the close association of these potential fungal endophytes with ecologically important phytoplankton groups.

## 4. Materials and Methods

### 4.1. K. veneficum Isolation, Maintenance, and Identification

A total of 13 strains initially isolated from 5 geographic origins were investigated in our study (Table 2 and Appendix A). The strain KV7+8, which is a mixture of the clonal cultures of KV7 and KV8, was isolated from an estuary of Chesapeake Bay, Virginia, the United States, in 2006. The other 12 strains were established from coastal waters in China, from the north to the south (Shandong, Fujian, Guangdong, and Guangxi provinces), from 2016 to 2020 (Table 2). The samples harvested for genomic DNA extraction (see below) were performed on December 20, 2020; thus, these cultures have been maintained in the laboratory for several months to over 14 years. All the cultures were kept in f/2 medium without Na_2_SiO_3_ [67], prepared with sterile filtered (0.22 µm membrane filter; Millipore, Billerica, MA, USA) natural seawater with a salinity of approximately 32–33 ppt. The cultures were maintained in an incubator at 20 ± 1 °C under a 12:12 h L/D cycle with 100 μmol photons m^−2^ s^−1^. We confirmed the identities of all the strains by high-throughput amplicon sequencing of the 28S rRNA genes (~500 bp, covering the highly variable D2 domain, and parts of the more conservative D1 and D3 domains) (BioProject: PRJNA824339; [36]).

### 4.2. Sample Collection and Genomic DNA Extraction

Cultures at the exponential growth stage were inoculated into six-well culture plates (Costar-Corning, New York, NY, USA) containing 10 mL of seawater-based f/2-Si medium and then incubated for 15 days under the same temperature and illumination as the routine culture conditions. Samples were then collected when all the cultures were at their stationary growth stage, as pre-determined. All the cells in each sample (approximately 10^4^~10^5^ cells for each sample) were pelleted in a 1.5 mL centrifuge tube and immediately used for genomic DNA isolation. Genomic DNA was extracted using the Plant DNA Extraction Kit (Tiangen, Beijing, China) according to the manufacturer’s protocols, and then it was suspended in 50 μL of TE buffer. Nuclear-free water processed through DNA extraction was used as the sample blank. The DNA was quantified using a NanoDrop^TM^ 1000 spectrophotometer (Thermo Fisher Scientific, Waltham, MA, USA) and stored at −80 °C until further use.

### 4.3. Gene Amplicon Sequencing of the rDNA ITS (Internal Transcribed Spacer) Region

The rDNA ITS2 region (~353 bp fragment) was amplified using primers fITS7 (5′-GTGARTCATCGAATCTTTG-3′) and ITS4 (5′-TCCTCCGCTTATTGATATGC-3′) [68] under the following conditions: 30 s at 98 °C; 35 cycles of denaturation at 98 °C for 10 s, annealing at 54 °C for 30 s, extension at 72 °C for 45 s; a final step at 72 °C for 10 min. The PCR reactions were conducted in a 25 μL mixture containing 12.5 μL of 2× Phusion^®^ Hot Start Flex Master Mix, 2.5 μL of each primer (1 μM), and 50 ng of the template DNA. The 5′ ends of the primers were tagged with specific barcodes per sample and subsequently yielded paired-end reads to each sample based on their unique barcodes. Nuclease-free water served as the blank. The resulting amplicons were collected and purified using a Gel Extraction Kit (Axygen Biosciences, Union City, CA, USA). The size and quantity of the purified amplicon libraries were assessed on an Agilent 2100 Bioanalyzer (Agilent, Palo Alto, CA, USA) before they were pyrosequenced on the NovaSeq PE250 platform (LC-Bio Technology Company, Hangzhou, China).

### 4.4. Sequencing Data Processing, Bioinformatic Analyses, and Fungal Metabolic Functional Prediction

Quality control of the raw data was performed with Fqtrim software (version 0.94), and chimeric reads were further excluded using the Vsearch tool (version 2.3.4) [69]. The paired-end reads were demultiplexed and assigned to respective samples according to their unique barcodes and then merged using FLASH [70]. The amplicon sequence variants (ASVs, sequences clustered at 100% sequence similarity) were denoised and yielded with the DADA2 package [71]. We assigned the taxonomy to the ASVs using the consensus blast method of the “feature-classifier” plugin in QIIME 2 [72] against the UNITE dynamic database [73,74]. Relative abundance of each ASV was estimated based on its read counts normalized to the total number of good quality reads. A heatmap was generated using the PHYLOTEMP tool, with relative abundance data clustered based on the Bray–Curtis similarity algorithm. Alpha diversity metrics (Shannon diversity, Simpson evenness, Chao1 richness, observed ASVs, and goods coverage) were calculated using the qiime2-diversity-alpha method. Beta diversity of the PCoA (principal coordinate analysis) was conducted with the QIIME 2 plugin based on the weighted UniFrac distance [72]. The functional profiles of the fungal communities were predicted using the PICRUSt algorithm [75] to draw inferences from the MetaCyc metabolic pathway database [76]. The output of PICRUSt2 was visualized and statistically analyzed with STAMP version 2.1.3 [77].

## 5. Conclusions

In this study, we used high-throughput amplicon sequencing to unveil the community composition and structure of the fungal consortium in coexistence with laboratory-reared marine dinoflagellate *K. veneficum*. The fungal consortium displayed remarkably high diversity and was dominated by filamentous fungi and ascomycetous and basidiomycetous yeasts, with features covering cosmopolitan, cold-adapted, and thermophilic taxa that have been ecologically reported as saprobic, mutualistic, or parasitic species. The fungal taxa, previously documented as endophytes in other hosts, were universally prevalent in all the *K. veneficum* culture samples, which implies that dinoflagellates potentially serve as an alternative ecological niche for fungal endophytes. The findings presented here will enrich our current knowledge on the specific associations of fungal consortia with marine dinoflagellates, and inspire further investigations into the potential functions exerted by these fungal associates during marine dinoflagellate blooms.

## Figures and Tables

**Figure 1 ijms-24-01672-f001:**
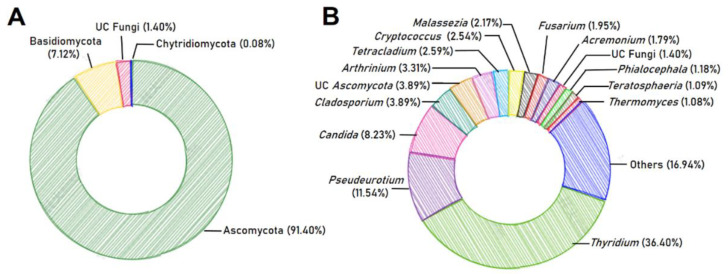
Taxonomic composition of the whole fungal microbiome coexisting with *Karlodinium veneficum* at (**A**) phylum and (**B**) genus levels. Pie charts represent the relative abundance of phyla/genera for the entire fungal community. UC: unclassified.

**Figure 2 ijms-24-01672-f002:**
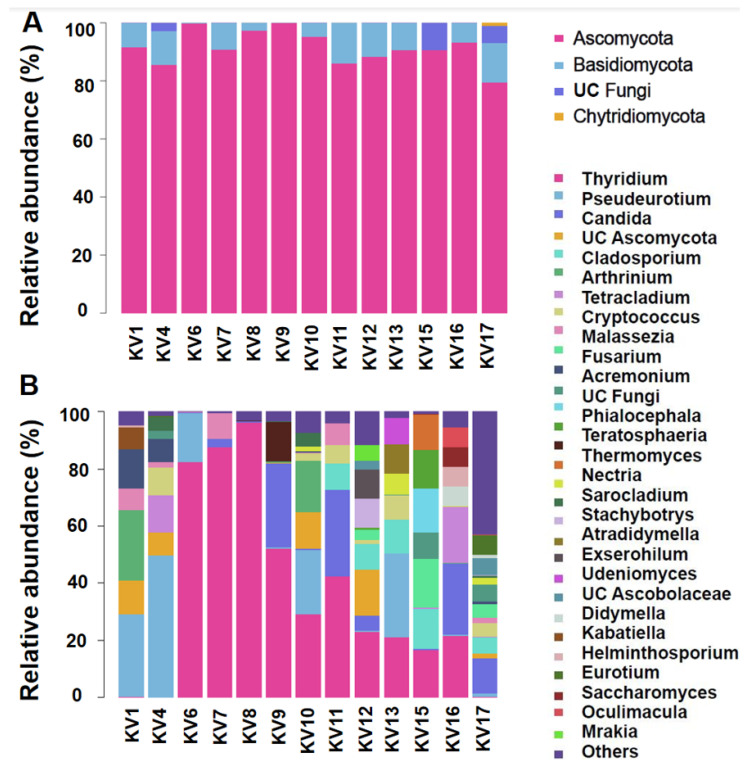
Relative abundance of fungi at (**A**) phylum and (**B**) genus (top 30) levels. Abundance is presented as a percentage of total effective ASVs in the samples. UC: unclassified.

**Figure 3 ijms-24-01672-f003:**
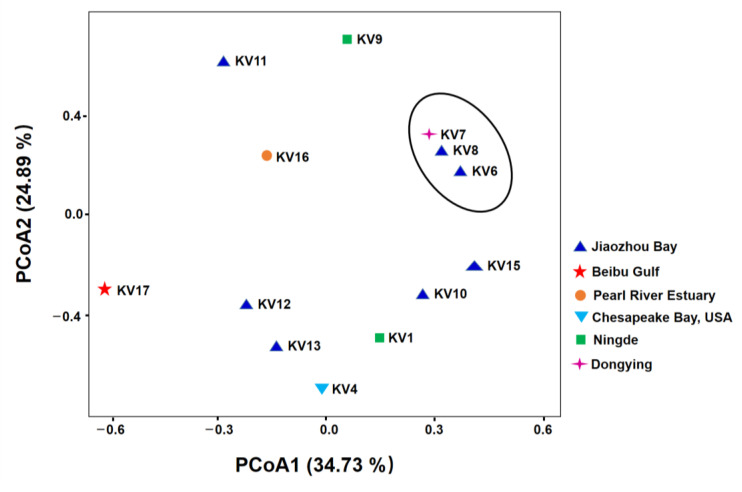
Principal coordinate analysis (PCoA) of fungal communities based on weighted UniFrac distances. The colors of the symbols indicate the original isolation sources of the samples.

**Figure 4 ijms-24-01672-f004:**
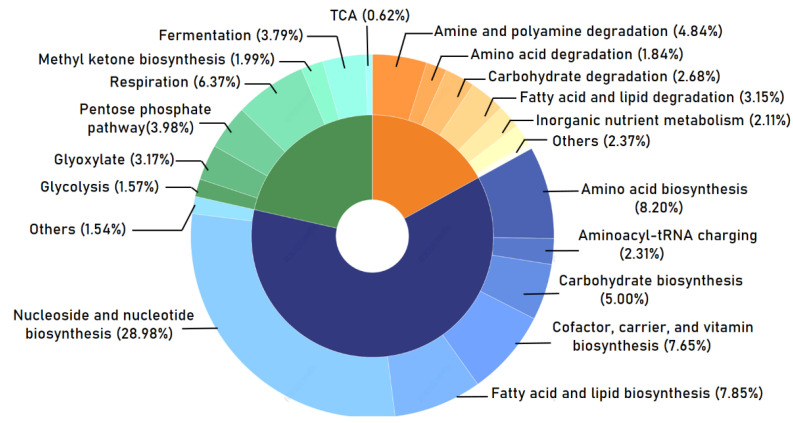
Functional assignments of the entire fungal microbiome coexist with *K. veneficum.* Gene functions were predicted from rDNA ITS region-based fungal compositions using the PICRUSt algorithm to draw inferences from the MetaCyc metabolic pathway database. Relative signal intensity normalized by the number of genes for each indicated metabolic pathway. The whole fungal microbiome generally performs the broad functions of biosynthesis (61.52%; blue), degradation/utilization/assimilation (16.99%; orange), and the generation of precursor metabolites and energy (21.49%; green).

**Table 1 ijms-24-01672-t001:** Occurrence and abundance of 15 abundant genera (relative abundance > 1% in all samples as a whole) in 13 samples.

Sample ID	Members
KV1	*Pseudeurotium* ^b^; *Arthrinium* ^b^; *Acremonium* ^c^; *Cladosporium* ^c^; *Malassezia* ^d^; *Thyridium* ^e^; *Candida* ^e^; *Fusarium* ^f^; *Cryptococcus* ^f^; UC Fungi ^f^
KV4	*Pseudeurotium* ^b^; *Tetracladium* ^c^; *Cryptococcus* ^d^; *Cladosporium* ^d^; *Acremonium* ^d^; UC Fungi ^d^; *Malassezia* ^d^; *Thyridium* ^e^; *Candida* ^e^
KV6	*Thyridium* ^a^; *Pseudeurotium* ^c^; *Candida* ^e^; *Malassezia* ^e^
KV7	*Thyridium* ^a^; *Candida* ^d^; *Malassezia* ^d^; *Pseudeurotium* ^e^; UC Ascomycota ^f^; *Fusarium* ^f^
KV8	*Thyridium* ^a^; *Pseudeurotium* ^e^; *Candida* ^e^; *Thermomyces* ^e^; *Cladosporium* ^f^; *Arthrinium* ^f^; *Tetracladium* ^f^
KV9	*Thyridium* ^a^; *Candida* ^b^; *Thermomyces* ^c^; *Pseudeurotium* ^e^; *Cladosporium* ^e^; *Arthrinium* ^e^; *Cryptococcus* ^f^; *Malassezia* ^f^; UC Ascomycota ^f^; *Acremonium* ^f^
KV10	*Thyridium* ^b^; *Pseudeurotium* ^b^; *Arthrinium* ^c^; *Cladosporium* ^c^; *Cryptococcus* ^d^; *Candida* ^e^; *Acremonium* ^e^; UC Ascomycota ^e^; *Malassezia* ^e^
KV11	*Thyridium* ^b^; *Candida* ^b^; UC Ascomycota ^d^; *Malassezia* ^d^; *Cryptococcus* ^d^; *Pseudeurotium* ^e^; *Arthrinium* ^f^
KV12	*Thyridium* ^b^; *Cladosporium* ^c^; UC Ascomycota ^d^; *Candida* ^d^; *Fusarium* ^d^; *Cryptococcus* ^d^; *Teratosphaeria* ^e^; *Pseudeurotium* ^e^
KV13	*Thyridium* ^b^; *Pseudeurotium* ^b^; UC Ascomycota ^c^; *Cryptococcus* ^d^; *Candida* ^e^; *Fusarium* ^f^; *Phialocephala* ^f^; *Teratosphaeria* ^f^; UC Fungi ^f^
KV15	*Thyridium* ^c^, *Fusarium* ^c^; *Phialocephala* ^c^; UC Ascomycota ^c^; *Teratosphaeria* ^c^; UC Fungi ^d^; *Pseudeurotium* ^e^; *Candida* ^e^; *Tetracladium* ^e^
KV16	*Thyridium* ^b^; *Candida* ^b^; *Tetracladium* ^c^; *Pseudeurotium* ^e^; *Cladosporium* ^f^; UC Ascomycota ^f^; *Arthrinium* ^f^; *Acremonium* ^f^
KV17	*Candida* ^c^; UC Fungi ^d^; UC Ascomycota ^d^; *Fusarium* ^d^; *Cryptococcus* ^d^; *Malassezia* ^d^; *Cladosporium* ^d^; *Thyridium* ^d^; *Acremonium* ^d^; Tetracladium ^e^; *Thyridium* ^e^

Note: Superscript letters denote relative abundance of genera in this sample (a: ≥50%; b: 20–50%; c: 10–20%; d: 1–10%; e: 0.1–1%; f: ≤0.1%). UC: unclassified.

**Table 2 ijms-24-01672-t002:** Details of 13 *K. veneficum* strains.

Sample ID	Strain Number	Origin	Isolation Date
KV1	KVND-1-cyst	Ningde, Fujian province (East China Sea)	Cyst germination, 2016
KV4	KV7+8	Chesapeake Bay, Virginia, United States	Vegetative cell, 2006
KV6	KVJZBXG01	Jiaozhou Bay, Shandong province (Yellow Sea in China)	Vegetative cell, 2019
KV7	DYKV7	Dongying, Shandong province (Bohai Sea in China)	Vegetative cell, 2020
KV8	KVJZBXG2020-8	Jiaozhou Bay, Shandong province (Yellow Sea in China)	Vegetative cell, 2020
KV9	I-K1-KV	Ningde, Fujian province (East China Sea)	Vegetative cell, 2019
KV10	KVJZBXG04	Jiaozhou Bay, Shandong province (Yellow Sea in China)	Vegetative cell, 2019
KV11	KVJZBXG05	Jiaozhou Bay, Shandong province (Yellow Sea in China)	Vegetative cell, 2019
KV12	KVJZBXG06	Jiaozhou Bay, Shandong province (Yellow Sea in China)	Vegetative cell, 2019
KV13	KVJZBXG07	Jiaozhou Bay, Shandong province (Yellow Sea in China)	Vegetative cell, 2019
KV15	KVJZBXG09	Jiaozhou Bay, Shandong province (Yellow Sea in China)	Vegetative cell, 2019
KV16	KVPRE1	Pearl River Estuary, Guangdong province (South China Sea)	Vegetative cell, 2019
KV17	P4-1	Beibu Gulf, Guangxi province (South China Sea)	Vegetative cell, 2018

## Data Availability

Data are contained within the article and Appendix A.

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
