# Peer review of "Different Geographic Strains of Dinoflagellate Karlodinium veneficum Host Highly Diverse Fungal Community and Potentially Serve as Possible Niche for Colonization of Fungal Endophytes"

_ijms, 2023, doi:10.3390/ijms24021672_

Round 1

Reviewer 1 Report (Previous Reviewer 1)

Dear Authors, I have reviewed this manuscript again and unfortunately I still think that the results presented are poorly supported by analyses: the lack of adequate replication and consequently, the lack of any statistical analysis. The authors conducted the analysis based only single samples of the dinoflagellate Karlodinium veneficum collected from 5 geographic locations. The only way to get correct results is to increase the level of replication. In my opinion, more samples and replicates are needed to verify endophytes of fungi identified in Karlodinium veneficum cultures.

Since there is such a large difference between the KV17 sample and others the addition of replicates could lead us to more convincing of the sequencing results. The Authors ‘speculate that this large difference might be simply caused by the difference in fungal diversity and abundance in the original seawaters’. Again, a deeper analysis will be helpful in solving this problem.

Moreover, I still feel that the data from natural populations in the field would be really helpful in improving this study and comparing detected core/unique/related with environment fungal members, as well as gaining insight into the pattern of fungal dynamics. The Authors wrote that ‘To gain insight into the pattern of fungal dynamics in natural seawater, techniques like the fluorescence/confocal microscopy observation with specific staining, and CARD-FISH with specific probes for in situ investigation of marine fungi will certainly be applied to our future studies.’ I propose to add these studies to this manuscript. Please also note that the negative control in the experiment is significant.

Author Response

Reviewer 2 Report (New Reviewer)

Overall, this is an interesting study about the association of fungi with dinoflagellates. It is rigorously conducted and reports novel findings. It fits well in the remit of this journal.  The quality of the figures is good and appropriate.

I have made numerous comments on the attached pdf.

However, the use of English in the paper is very poor and I would request that the paper is revised by a native English speaker or a professional language editing service for scientific texts (I would e.g. recommend Arete Writing  https://www.arete-writing.com/   - but there are certainly numerous others). A revised version should not be sent out for peer review again unless the authors can prove that the text has been revised either by a native English speaker or a professional language editing service.

Author Response

This manuscript is a resubmission of an earlier submission. The following is a list of the peer review reports and author responses from that submission.

Round 1

Reviewer 1 Report

This manuscript presents the results of a fungal community study of 13 clonal cultures of the dinoflagellate Karlodinium veneficum collected from 5 geographic locations and maintained under laboratory conditions. The authors demonstrated that the dinoflagellate was colonized by a fungal endophyte. The results are interesting, however, I think they are poorly supported by analyses, since only single samples of different geographic strains were sequenced. There were huge differences in the number of ASVs obtained ranging from 23 to 120. Nevertheless, based on the single data, the authors could not be sure whether this was due to contamination or some other reason. The second major doubt relates to the described “different geographic strains” – how different are they? Phylogenetic analysis is required. Last, but not least, a major suggestion regarding the unexpected fungal endophytes in the marine environment, as described by the authors – more evidence should be provided, i.e. additional molecular marker should be amplified or phylogeny analysis based on the obtained ITS2 fragment sequences should be conducted involving similar sequences available in the GenBank database.

Introduction

Line 67-68: I wonder how precise ASVs for marine fungi can be obtained since they have been poorly studied, and as a result, many references are not available in databases. Are you sure  PICRUSRt analyses are correct then?

Line 114: references are needed.

Results

Please provide the alpha rarefaction curve.

Why there is such a large difference between the KV17 sample and others, i.e., the Chao1 richness for KV17 was 120 and this value for all the others samples ranged from 23 to 63. I think that additional sequencing is needed to exclude potential errors.

Line 166 – Figure 1 is unreadable and I am not convinced that it presents the whole fungal microbiome living in K. veneficum, since there were huge differences between the samples that were sequenced without replicates. I understand that the presented work is a continuation of the study of Deng et al. (2022), and I believe that the authors did all they could, but this result is highly ambiguous.

Deng, Y.Y.; Wang, K.; Hu, Z.X.; Tang, Y.Z. Identification and implications of a core bacterial microbiome in clonal cultures laboratory-reared for months to years of the cosmopolitan dinoflagellate Karlodinium veneficum. Front. Microbiol. 2022, 13, 967610

The quality of all figures presented should be improved.

Please also provide the list of clusters of the obtained sequences to ASVs.

Discussion

Line 226 and 240 references are needed.

The authors should provide sequences of all described fungal endophytes and a newly described genus Sordariomycetes, and for all of them, a comparison with available sequences deposited in GenBank, as well as phylogeny analysis, should also be performed.

Line 292-294 This is a good point, but without samples from the natural environment, the authors cannot be certain if there is any contamination. e.g. from the laboratory.

Materials and methods

Line 335 Why were strains KV7 and KV8 mixed?

Unfortunately, I feel that the data from natural populations in the field would be really helpful in improving this study and comparing detected core/unique/related with environment fungal members, as well as gaining insight into the pattern of fungal dynamics.

Reviewer 2 Report

This study describes the composition of fungi by metabarcoding of 13 cultures of the dinoflagellate Karlodinium veneficum.

In the last years, numerous Chinese labs have the instrument and software for metabarcoding, and they have to amortize the investment. Sometimes with meaningless studies as it is in this case.

During years, I have carried molecular studies of dinoflagellates using single cell PCR, often with the odious results of amplifying the sequence of a fungus. Yes, the fungi with their spores are everywhere. When you are in autumn in France, you obtain sequences of the mushrooms that grow in the garden outside the lab instead of the isolates of dinoflagellates collected in the central Pacific Ocean at 100 m depth. When you have cultures of dinoflagellates, even if in salty waters, the spores that are in the air are able to grow and feed with the organic matter available in the culture. It is common to place a Petri dish in the lab with some culture media, and to check periodically for the presence of fungi that will be a source of potential contaminants.

In this study, the 99% results that the authors have obtained are unrelated to the dinoflagellate Karlodinium veneficum. If you analyze the composition of fungi in the cotton bud after placing it in the hollow of your belly button, you will obtain the same results that in the culture of Karlodinium veneficum. The 99% of the results in this study are contaminants, fully unelated Karlodinium veneficum. Probably the only interesting result is the presence of Chytridiomycota in the strain KV17.

The results of the experiment are unacceptable because there are no a controls. There is no any comparison between the seawater/culture media with and without dinoflagellate cells, or with any other microalgae. There is no any study of the composition of fungi in the surrounding the water where the cells were isolated, or in the initial stage of the culture before the contamination with external fungi. There is no a study of the evolution of the composition of fungi in the cultures over time. There is no a study of the relative composition of fungi in the distinct stages of the culture (exponential growth, senescence, etc.)

The scarce scientific value is to know the fungi that have contaminated the cultures, but without a time-series or without comparison cultures of distinct species and without microalgae, there is nothing that deserves to be published.

I cannot recommend the publication of this manuscript. I recommend to the authors to improve the design of the experiments in order to avoid to waste public money.

Minor comments

Abstract line 29: more than 14 years were investigated in this study      

Please report your results of metabarcoding that you obtained 14 years ago.

Abstract line 42: the possibility that dinoflagellates can serve as an ecological niche for the colonization of  fungal endophytes

Metabarcoding analysis of a filter is not the scientific procedure. If you want to see the fungi inside the dinoflagellate, just observe them with a microscope. There are distinct methods such as fluorescence/confocal microscopy after specific staining, or serial cuts of the cell by transmission electron microscopy. Like this, you can see the fungus inside the cell, and later you can design an experiment to obtain molecular data of the endobiont or prey of the dinoflagellate.

page 2, line 62: evidences have shown that the mutual interactions between phytoplankton and microbiota act as a vital factor influencing the dynamics of  both algal populations and microbial communities [3-13].

It is not needed to fill out the text with many references [3-13]. This manuscript is not about the mutual interactions between phytoplankton and microbiota. This is about the fungi that are in the air of the lab, and they have contaminated the cultures.

line 42: Dinoflagellates constitute the most cosmopolitan and diverse phytoplankton in coastal marine ecosystems

Not, diatoms constitute the most cosmopolitan and diverse phytoplankton in coastal marine ecosystems

line 119: 119 zooplanktons

zooplankton is already a plural word.

line 220:   3. Discussion

3.1. K. veneficum harbor fungal communities with remarkably high diversity

Please replace the subtitle by "The cultures of K. veneficum and our lab were contaminated with remarkably high diversity of fungi”

line 289: flagellates (e.g. Kareniacea),

Do you mean the family Kareniaceae? The correct name for this family of dinoflagellates is Brachidiniaceae.

line 340: NaSiO3

Na2SiO3?

line 342: 0.22 mm membrane filter

There are not filters of 0.22 mm. The filter are 10, 25, 33 mm or other diameter sizes.

There are not filters with a pore size of 0.22 mm.

You use a 0.22 µm pore size membrane filter.

line 351 Corning, US)  What country is US?   United States of Mexico?

line 354: 104~105 cells

What means?

line 361: 4.3. ITS (internal transcribed spacer) rRNA gene amplicon sequence

As molecular biologist you should know that the ITS region is not ribosomal RNA (rRNA)

line 367: The 5’  ends of the primers were tagged with specific barcodes per sample and sequenced using universal primers.

That explains nothing.

Round 2

Reviewer 2 Report

This manuscript reports the sequences of fungi found in the seawater used for culturing several strains of a dinoflagellate. As reported in the first revision, nearly all the fungi found in the cultures are from aerial contamination. The same fungi that are in the bench of the laboratory are in the seawater of your cultures. Consequently, these fungi can be found in these cultures of dinoflagellates, in cultures of a freshwater chlorophyta, or everywhere. It is wrong to associate these sequences of fungi with this species of dinoflagellates. There is no any sample use as negative control in the experiment.

No serious journal will accept the publication, except predatory journals. I cannot waste my time with this kind of manuscripts.

Just a note: The sequence of the type species of Brachidinium Taylor 1963 branched among the sequences of Karenia Mostreup & Hansen 2000. Then, both Brachidinium and Karenia must be classified in the same family. We have the family Brachidiniaceae Sournia 1972 and the family Kareniaceae Bergholtz et al. 2005. The article 11.3 of  International Code of Nomenclature established: For any taxon from family to genus, inclusive, the correct name is the earliest legitimate one with the  same rank. 

The family Brachidiniaceae Sournia 1972 has the priority over Kareniaceae Bergholtz et al. 2005. No discussion is possible: Brachidiniaceae is the correct name for this family of dinoflagellates. Please do not believe everything that you found in Internet.  A scientist is based on scientific facts, and not in something that someone wrote in Internet.